# CLIC1 Protein Accumulates in Circulating Monocyte Membrane during Neurodegeneration

**DOI:** 10.3390/ijms21041484

**Published:** 2020-02-21

**Authors:** Valentina Carlini, Ivan Verduci, Francesca Cianci, Gaetano Cannavale, Chiara Fenoglio, Daniela Galimberti, Michele Mazzanti

**Affiliations:** 1Dipartimento di Bioscienze, Università degli Studi di Milano, 20133 Milano, Italy; valentina.carlini@unimi.it (V.C.); ivan.verduci@unimi.it (I.V.); francesca.cianci@unimi.it (F.C.); gaetano.cannavale@unimi.it (G.C.); 2Dipartimento di Fisiopatologia medico-chirurgica e dei trapianti, 20122 Milano, Italy; chiara.fenoglio@unimi.it; 3Dipartimento di Scienze biomediche, chirurgiche e odontoiatriche, 20122 Milano, Italy; 4Fondazione IRCCS Ca’ Granda, Ospedale Maggiore Policlinico, 20122 Milano, Italy

**Keywords:** neurodegeneration, CLIC1 protein, PBMC, monocytes, cell membrane, chloride channels

## Abstract

Pathologies that lead to neurodegeneration in the central nervous system (CNS) represent a major contemporary medical challenge. Neurodegenerative processes, like those that occur in Alzheimer’s disease (AD) are progressive, and at the moment, they are unstoppable. Not only is an adequate therapy missing but diagnosis is also extremely complicated. The most reliable method is the measurement of beta amyloid and tau peptides concentration in the cerebrospinal fluid (CSF). However, collecting liquid samples from the CNS is an invasive procedure, thus it is not suitable for a large-scale prevention program. Ideally, blood testing is the most manageable and appropriate diagnostic procedure for a massive population screening. Recently, a few candidates, including proteins or microRNAs present in plasma/serum have been identified. The aim of the present work is to propose the chloride intracellular channel 1 (CLIC1) protein as a potential marker of neurodegenerative processes. CLIC1 protein accumulates in peripheral blood mononuclear cells (PBMCs), and increases drastically when the CNS is in a chronic inflammatory state. In AD patients, both immunolocalization and mRNA quantification are able to show the behavior of CLIC1 during a persistent inflammatory state of the CNS. In particular, confocal microscopy analysis and electrophysiological measurements highlight the significant presence of transmembrane CLIC1 (tmCLIC1) in PBMCs from AD patients. Recent investigations suggest that tmCLIC1 has a very specific role. This provides an opportunity to use blood tests and conventional technologies to discriminate between healthy individuals and patients with ongoing neurodegenerative processes.

## 1. Introduction

The number of cases of neurodegenerative diseases is growing constantly in our society, mainly due to the increase in life expectancy. For example, in sporadic Alzheimer’s disease (AD), the pathological phenotype occurs between 65 and 70 years old, with an exponential increase in incidence with population aging. Delaying the start of the disease by about five years would be enough to reduce medical costs by around 50% [1]. At present, diagnostic procedures are difficult and not always reliable. Furthermore, in most cases, diagnosis occurs when neurodegenerative processes are in an advanced state and when the cognitive and physical abilities of the patient are already compromised. Therefore, the real origin(s) of the disease is/are difficult to determine. Current therapies are based on anti-inflammatory and antioxidant drugs or sedatives. Recent studies and clinical trials have demonstrated that these current therapies are unable to stop the neurodegenerative process, and also that is still not possible to decrease the incidence of the disease [2,3]. There is scientific consensus on the idea that identification of early markers of the neurodegenerative process could pave the way for better clinical trials, including the selection of patients who are not yet irreversibly compromised. At the moment, the most advanced and reliable diagnostic tools are the quantification of free beta amyloid and tau peptides in the cerebrospinal fluid (CSF) [4,5]. Although the test has good accuracy (about 90% sensitivity and specificity), this methodology is invasive, hence it is not suitable for a large-scale population screening. The measurement of other proteins in CFS, such as neurofilament light chain (NFL) and of microRNAs in plasma/serum, imaging techniques, and assays based on protein analysis are also available but less commonly used [6]. However, imaging techniques are expensive and RNA-protein analyses require adequate equipment and expertise to be routinely used in clinical practice. Therefore, there has been an intense search for additional neurodegenerative biomarkers in blood [4,5,7]. Our work aims to define the chloride intracellular channel 1 (CLIC1) protein localization as a marker of the chronic inflammatory state of the central nervous system (CNS). CLIC1 belongs in the category of metamorphic proteins [8,9]. It can be found as a hydrophilic soluble element as well as an integral membrane protein (tmCLIC1). Once in the transmembrane form, CLIC1 is associated with chloride ions’ permeability [10,11]. Recent investigation demonstrated that tmCLIC1 and NADPH oxidase are involved in a feed-forward mechanism that increases the production of reactive oxygen species (ROS) [12], particularly in macrophages [13]. This is instrumental in speeding up the transition G1/S phase during the cell cycle [14]. Furthermore, it has been demonstrated that CLIC1 is involved in axonal outgrowth [15], suggesting a functional role of the protein in relation to the cytoskeleton [16,17]. More recently, the presence of CLIC1 has been shown on the cleavage furrow [16], which confirms there is a close relationship between CLIC1 and cell membrane plasticity. The role of CLIC1 (and CLIC4) has been described in mouse macrophages in response to LPS stimulation. The two proteins accumulate in the plasma membrane and in the nucleus where they are involved in the production of IL-1*β* [18]. Previous investigation has shown tmCLIC1 protein accumulation in beta-amyloid-stimulated microglia cells [19]. CLIC1 chloride current inhibition or protein downregulation showed a neuroprotective effect in neuron-microglia rat co-culture in the presence of beta amyloid peptide [19]. It has been demonstrated that chronic inflammatory states of the CNS, like the neurodegenerative process, activate microglia cells and peripheral blood monocytes. Activated monocytes are recruited in the CNS through the blood brain barrier (BBB) to become tissue-resident macrophages [20,21,22,23,24,25,26]. Our hypothesis is based on the idea that chronically-activated PBMCs, and in particular monocytes, accumulate tmCLIC1 protein. Furthermore, CLIC1’s peculiar localization could be functional to monocytes’ proliferation and infiltration [27]. In the present investigation, we demonstrate that there is an increased expression of tmCLIC1 protein in the monocytes of confirmed AD patients compared to elderly healthy individuals. Conclusive data were collected using immunolocalization with an antibody directed against the external portion of the protein [12], and electrophysiological recordings of the membrane current [12,19]. 

## 2. Results

### 2.1. CLIC1 Protein Increases in AD Patient Monocytes

The goal of our research was to validate CLIC1 from circulating monocytes as a potential marker for a progressive state of the neurodegenerative process. We selected patients with a clear diagnosis of AD to reduce variability. We also decided to only consider monocytes in immunofluorescent experiments because they show a more intense CLIC1 signal compared with lymphocytes (Appendix A). 

Human monocytes isolated from peripheral blood obtained from young (*n* = 20) and old (*n* = 27) healthy individuals and from AD patients (*n* = 25) were fixed and immunostained for CLIC1 protein using a whole protein commercial antibody. Average fluorescence plots (Figure 1 and Appendix A) show a marked increase in the presence of CLIC1 in monocytes from AD patients’ blood. 

The assessment of CLIC1 overexpression in circulating monocytes as a marker for the presence of a neurodegenerative process in the CNS implies the identification of a reliable and easily quantifiable parameter. Several alternatives can be verified. To confirm the increase in the presence of CLIC1 in AD patients’ circulating monocytes, mRNA was collected from the blood samples of 29 elderly individuals (11 males and 18 females) and 35 AD patients (16 males and 19 females) (see Materials and Methods section).

Figure 2 depicts the quantification of CLIC1 protein mRNA from control elderly individuals and AD patients stratified by gender (Figure 2a) and together (Figure 2b). Although the measurements show a marked increase in CLIC1 mRNA in AD patients, there is still overlapping data between the two populations.

Similar results can be obtained from the ratio between CLIC1 fluorescence in the cytoplasm and in the nucleus of the cells. Figure 3 shows that the ratio in AD patients is significantly higher compared to healthy subjects. However, as for mRNA quantification, the cytoplasmic/nucleoplasm fluorescence ratio generates an unpredictable number of false positive cases. 

### 2.2. CLIC1 Protein Membrane Localization as a Neurodegenerative Marker 

CLIC1 protein localization might be a better way to discriminate physiological and pathological monocytes. Previous investigations have demonstrated the translocation of CLIC1 protein during microglia activation [12,19]. In homeostatic conditions, CLIC1 is a cytoplasmic, hydrophilic protein. Unbalanced environment, like chronic oxidative stress, drives CLIC1 protein towards the plasma membrane. After structural rearrangement and dimerization, CLIC1 becomes a transmembrane protein that is able to promote selective chloride permeability [8,9]. CLIC1 membrane insertion was quantified using a polyclonal antibody directed against the amino terminus protein domain exposed on the extracellular side of the membrane (Appendix A) [12]. The fluorescent signal was analyzed within the selected membrane area (Appendix A). 

Figure 4a shows confocal microscope images of monocytes derived from healthy subjects and AD patients. Freshly isolated cells were incubated with CLIC1 amino-terminal antibody before fixation to avoid possible membrane damage that would also result in the staining of cytoplasmic CLIC1 (Appendix A). 

A chart plot of the data from control and AD patients (Figure 4b) demonstrates adequate separation of the membrane fluorescence measurements. These last results confirm our hypothesis. CLIC1 protein not only increases during progressive neurodegenerative processes, but it also correlates with enrichment at the plasma membrane level. The striking difference between a healthy and a neuropathological condition of the CNS is the amount of protein that is able to colonize the plasma membrane. Activation of peripheral monocytes induced by microglia signals diffused by the blood stream starts the process of monocytes’ transformation in macrophages. CLIC1 plays an active part in this development. According to recent investigations, membrane CLIC1 protein is involved in cellular proliferation, migration and infiltration by increasing cytosol oxidation and pH alkalization [12,14,19]. All these activities are promoted by chloride permeability, which has been associated with the presence of CLIC1 protein in the membrane in several studies. Furthermore, electrophysiological measurement is the most precise way quantify the CLIC1 translocation to the membrane. The membrane ionic permeability of isolated monocytes was recorded using a whole-cell perforated patch procedure. To isolate CLIC1 current, we used 100 µM indanyloxyacetic acid (IAA94), a chloride channel blocker that is specific for CLIC1 channels at this concentration. Figure 5 shows membrane ionic currents elicited by a 9-step voltage protocol (from −80 to +80 mV every 20 mV) from isolated monocytes obtained from healthy individuals (Figure 5a) and AD patients (Figure 5b). Average current is reported in a current/voltage relationship (Figure 5c) as current density (controls *n* = 5, AD patients *n* = 6). Both membrane protein localization and CLIC1-induced membrane current could be used to identify monocytes isolated from AD patients’ blood. 

## 3. Discussion

Neurodegeneration is a physiological process that occurs in all aging individuals. With the increase in life expectancy, it has become a major challenge for health care systems, particularly in Western countries. The most important parameter of neurodegeneration is the progression rate of neuronal death. It is commonly believed that there are diverse physical and environmental factors that speed up this process, as well as several factors that are able to delay neurodegeneration. These factors could be intrinsic or extrinsic, for example, the person’s specific metabolism and/or linked to the surrounding environment. Lifestyle is one of the major players in this process. Chronic stress on the central nervous system over several years, has a strong impact on cells’ aging, and in particular, on the neurodegenerative process. Food intake, for example, is considered a key factor. An unbalanced diet generates a constant state of oxidative stress that speeds up the degenerative process. A persistent imbalance between pro-oxidants and antioxidants towards the first, could cause a chronic state of inflammation. In the CNS, neurons are particularly vulnerable under oxidative stress conditions because they are not able to produce antioxidant compounds. A chronic inflammatory state caused by persistent oxidative stress conditions within the CNS is, nowadays, impossible to detect in the absence of a pathological phenotype. Most of the time, when cognitive impairment and physical problems occur, the progression of the neurodegeneration is too advanced for any successful therapeutic action. This scenario justifies all ongoing investigations that aim to identify blood biomarkers. The availability of an easy diagnostic test will allow the screening of a high number of aging individuals and detection of any CNS inflammatory condition at the beginning of the neurodegeneration process. Several studies have suggested the use of blood markers for the diagnosis of neurodegenerative disease [4,7,28,29,30]. Recently, Schipke and colleagues proposed the combined measurement of six different blood markers, BDNF, IGF-1, VEGF, TGF-beta 1, MCP-1 and IL-18 to identify Alzheimer’s disease patients [31]. The present investigation proposes CLIC1 as an additional blood indicator for a state of chronic inflammation. As a first approach, with the intent of reducing variability, we collected blood samples only from patients with advanced AD (see Methods). The blood samples from confirmed Alzheimer’s patients represent an extreme condition. However, there are several reasons for this choice. AD represents more than 70% of all neurodegenerative cases. Thus, there are a sufficient number of blood samples to conduct a population study. Moreover, there are enough different diagnostic tools, although most of them are invasive, to ensure sufficient homogeneity between subjects [6]. In this scenario, the detection of a protein that shuttles from the cytoplasm to the membrane only in the presence of chronic inflammatory conditions is worth investigating. CLIC1 protein is involved in several cell functions. It is an active element that contributes to the condition of oxidative stress in the cell [12,13,14], has functional connection with the cytoskeleton [15,16,17], and once in the membrane, it controls chloride flux through the lipid bilayer [10,11,19]. All these functions are linked to cell proliferation, cell migration and infiltration [11,13,15,16,17,19]. We focused on monocytes because they are the ideal elements to investigate CLIC1 properties. The activation of monocytes include proliferation, migration and local infiltration. In addition, the CLIC1 signal in immunofluorescence experiments is more pronounced in monocytes rather than in lymphocytes (Appendix A). In healthy conditions, monocytes are devoted to surveillance, and work as sensors of any abnormalities occurring in the different parts of the body. During infections or chronic stress conditions, monocytes are able to sense alert signals, be activated and contribute to the inflammatory process. CLIC1 protein plays an active role in the transformation of monocytes in resident macrophages. During the neurodegenerative process, systemic monocytes are recruited in the brain [20,24,26,27,32] following chemical messages sent in the blood stream by activated microglia [20,21,22,24,26]. Considering the properties of CLIC1, it is reasonable to think that it is involved in monocyte proliferation, infiltration into the brain parenchyma through the blood brain barrier, acquisition of macrophage characteristics and migration to the infection site. The main role of CLIC1 is to act as a metamorphic protein [8,9,19]. Translocation into the membrane allows CLIC1 to contribute to monocyte activation. PBMCs isolated from AD patients have shown CLIC1 mRNA overexpression. However, high CLIC1 mRNA content is not always concomitant with increased CLIC1 protein expression and function [33]. In addition, both CLIC1-fluorescence signal and mRNA levels present some gray zones between control subjects and AD patients. This would produce a number of false positive outcomes, which is not acceptable for a diagnostic tool. Because it has been previously demonstrated that the mechanism of cell activation is due to tmCLIC1 [12,14,15,19,33], we used a more accurate procedure based exclusively on the evaluation of the transmembrane form of the protein. As shown in Figure 4 and Figure 5, the techniques to obtain reliable measurements are more complex and present some drawbacks. However, confocal microscopy and patch-clamp techniques deserve more attention. In particular, patch-clamp, the most reliable way to quantify CLIC1 membrane function, requires specific instruments and it is highly time consuming. Even in the case of a computer-assisted multi-electrode patch-clamp system, the analysis is complex.

Future research perspectives include the development of a large-scale population screening based on isolated PBMCs. At the moment, the differences between isolated monocytes from controls and AD patients are evident because confocal microscopy and electrophysiology are very sensitive. The fluorescent signals coming from controls and AD cells are difficult to detect with conventional spectrophotometers. The objective is to increase the signal of the entire PBMCs population by using a better antibody or a more sensitive detection system. Alternatively, we could isolate only monocytes using a specific marker and then quantify tmCLIC1. We have shown that the protein accumulates preferentially in monocytes’ membrane. Although the number of monocytes is restricted, we are confident that it is possible to collect a sufficient number of cells from control individuals or from AD patients to obtain a useful comparison.

In conclusion, our investigation demonstrated that blood monocytes show membrane accumulation of CLIC1 protein in Alzheimer’s patients. To confirm tmCLIC1 as a reliable and specific marker for CNS neurodegeneration it will be important to also test the presence of this protein in other neurodegenerative processes, like frontotemporal dementia or Parkinson’s disease. It will also be important to test possible alteration of tmCLIC1 levels in systemic pathological states, like multiple sclerosis or peripheral neuropathy.

## 4. Materials and Methods

### 4.1. Population

Nineteen female and 16 male AD patients were recruited at the Alzheimer’s Center of the University of Milan, Fondazione Cà Granda, IRCCS Ospedale Maggiore Policlinico, between 2011 and 2014. All patients underwent a clinical interview, neurological and neuropsychological examination, routine blood tests, brain MRI and lumbar puncture (LP) for quantification of the CSF biomarkers A*β*, total tau and tau phosphorylated at position 181 (Ptau). The cut-off thresholds of normality were: A*β* ≥ 600 pg/mL; tau ≤ 500 pg/mL for individuals older than 70 years and ≤450 pg/mL for individuals aged between 50 and 70 years; Ptau ≤ 61 pg/mL. The diagnosis of AD was confirmed by their pathological CSF A*β* levels, according to the criteria of the International Working Group guidelines [34]. The Clinical Dementia Rating (CDR), the Mini Mental State Examination (MMSE), the Frontal Assessment Battery (FAB), the Wisconsin Card Sorting Test (WCST), and the Tower of London test assessed cognitive dysfunction. Individuals with significant vascular brain damage were excluded (Hachinski Ischemic Score <4). Eighteen female and 11 male age-matched and sex-matched controls (HC) were also recruited. All individuals underwent a LP to check for a CNS disease and were discharged with no evidence of neurological disease or cognitive impairment. All these subjects were followed up clinically and neuropsychologically for almost 3 years. None of them developed cognitive decline and at testing their MMSE was ≥ 28. To create the group of young controls, we selected twenty volunteers between 22 and 28 years old in good health. All the subjects were informed about the nature of the research and agreed to donate an 8 mL blood sample. The protocol was approved by the "Institutional Review Board of the Policlinico Hospital" (Neurology BS30-4673-2, 1 March 2011).

### 4.2. PBMCs Isolation

For each subject, 8 mL of venous blood was collected into BD Vacutainer^®^ CPT™ cell preparation tubes (BD Biosciences, Franklin Lakes, NJ, USA) containing 1 mL of Sodium Citrate and 2 mL of Ficoll. PBMCs were separated by gradient centrifugation according to manufacturer’s protocol. Freshly isolated PBMCs were immediately used for immunostaining or patch clamp experiments or frozen at −80 °C until RNA extraction.

### 4.3. Immunostaining

CLIC1 protein expression and cellular localization in freshly isolated PBMCs was analyzed through an indirect immunofluorescence protocol. For whole CLIC1 immunostaining, cells were fixed with 2% paraformaldehyde (SigmaAldrich, St Louis, MO, USA) and permeabilized with 0.05% Triton X-100 (SigmaAldrich), blocked in BSA 4% (SigmaAldrich) for 30 minutes at RT and incubated with primary mouse monoclonal anti-CLIC1 antibody (Santa Cruz Biotechnology, Dallas, TX, USA) for 1 h at RT. For transmembrane CLIC1 immunostaining, cells were directly blocked and incubated with a custom-made primary mouse polyclonal anti-NH2 CLIC1 antibody for 2 h on ice (Appendix A). The antibody, recognizing the first 24 amino acids facing the external membrane side, was provided by the laboratory of Dr. Samuel Breit from the University of New South Wales, Sydney, Australia [12]. After whole/NH2 CLIC1 antibody incubation, cells were incubated with fluorochrome-conjugated secondary antibodies (anti-mouse Alexa Fluor 488 and anti-mouse Alexa Fluor 555, Life Technologies, Carlsbad, CA, USA) for 1h at RT, counterstained with 200 ng/mL DAPI (SigmaAldrich) and mounted on microscope slides using a glycerol-based mounting medium. Samples were observed under a confocal microscope (Leica TCS SP2, Leica Microsystems, Wetzlar, Germany) with a Leica HCX PL APO 63 × /1.4 NA oil immersion objective. Images were analyzed using ImageJ software (U. S. National Institutes of Health, Bethesda, Maryland, USA).

### 4.4. Total mRNA Extraction and Quantitative Transcriptional Analysis by Real Time PCR

Total RNA samples were extracted from PBMCs using Trizol (Life Technologies), according to the single step acid phenol method. RNA purity was measured by optical density and only samples with an OD 260/280 ratio ranging between 1.8–2 and an OD 260/230 greater of equal to 1.8 were used. RNA was then reverse-transcribed by the Ready-To-Go You-Prime First-Strand Beads kit (Amersham Biosciences-GE Healthcare, Chicago, IL, USA). Quantitative analysis of CLIC 1 mRNA levels was performed using an ABI 7500 Sequence Detector with dual-labeled TaqMan probes. The relative amount of CLIC 1 mRNA was determined by the comparison of a specific TaqMan probe (Hs00559461_m1, Applied Biosystems, Foster City, CA, USA) with the housekeeping 18s rRNA probe (Hs99999901 s1, Applied Biosystems). RT-PCR amplification reactions were performed in a final volume of 20 µL, using the TaqMan Universal Master Mix (ABI 4324018, Applied Biosystems). The cycle parameters used were as follows: 2 min at 50 °C, 10 min at 95 °C, 40 cycles of 15 s at 95 °C for denaturation, 1 min at 60 °C for annealing/extension. Relative mRNA levels were calculated according to the ΔΔCt method, where ΔCt = Ct (analyzed gene) − Ct (housekeeping gene). Normalization was performed automatically by using the analysis software SDS v. 2.04 (ABI, Applied Biosystems).

### 4.5. Patch-Clamp Recordings

To measure CLIC 1-mediated current, the patch-clamp electrophysiology technique was performed. The cells were voltage-clamped in a perforated-patch whole-cell configuration as previously reported [15]. The voltage protocol consisted of 800 ms pulses from −80 mV to +80 mV (20 mV voltage steps). The holding potential was set according to the resting potential of the single cells. Analysis was performed using Clampfit 10.2 (Molecular Devices, Sunnyvale, CA, USA). CLIC 1-mediated chloride current was isolated from the other cell ionic currents by perfusing 100 μmol/L of the specific chloride channel inhibitor IAA94 (indanyloxyacetic acid 94, SigmaAldrich). The patch-clamp solutions used are as follows: bath (mmol/L): 125 NaCl, 5.5 KCl, 24 HEPES, 1 MgCl2, 0.5 CaCl2, 5 D-glucose, 10 NaOH, pH 7.4; pipette (mmol/L): 135 KCl, 10 HEPES, 10 NaCl, 1 MgCl2, pH 7.2.

### 4.6. Data Analysis and Statistics

Statistical analyses were performed with GraphPad Prism software 7 (GraphPad Software, Inc., San Diego, CA, USA). For analyses of two experimental groups, we used an unpaired two-tailed *t*-test. For comparing multiple groups, a one-way analysis of variance (ANOVA) followed by Tukey’s multiple comparisons test was performed. Differences between groups were considered significant at a *p* value of < 0.055.

Immunostaining experiments were performed as follows: for each subject, we randomly selected 50 single monocytes from 3 different 20× fields of a PMBCs dispersion. The average fluorescence for each subject has been plotted in every box chart of Figure 1, Figure 2, Figure 3 and Figure 4.

Electrophysiology data are reported as means ± SEM. CLIC 1-mediated current density/voltage relationships were analyzed by plotting the average of the last 50 ms current elicited current average at each test potential for control (*n* = 5) and AD patients (*n* = 6).

## Figures and Tables

**Figure 1 ijms-21-01484-f001:**
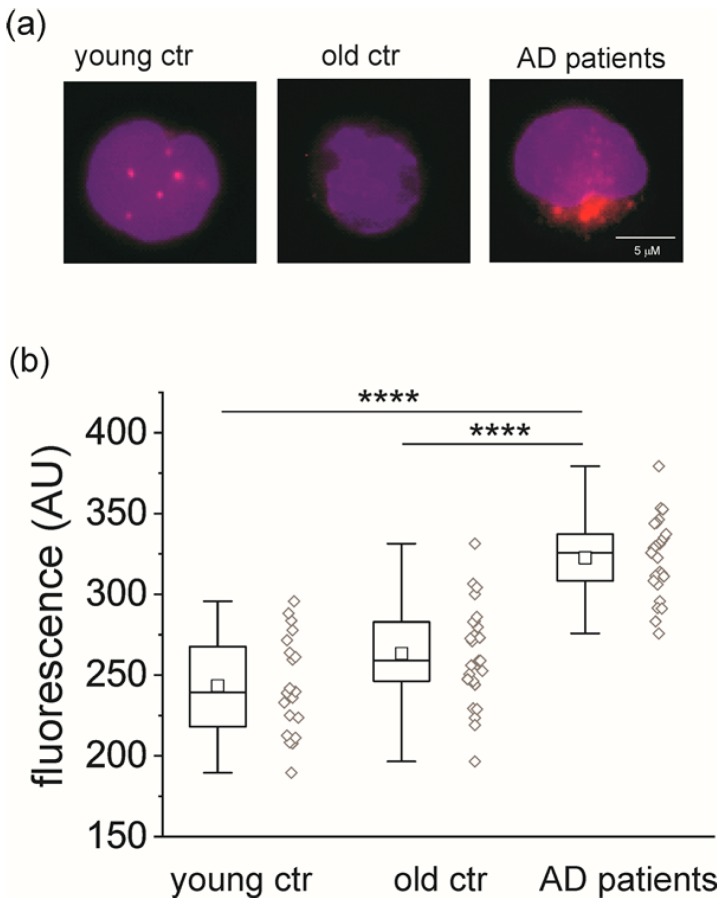
Chloride intracellular channel 1 (CLIC1) protein overexpression in circulating monocytes of Alzheimer’s disease (AD) patients. (**a**) Representative confocal images of CLIC1 protein staining (red) in monocytes (blue nuclei) isolated from peripheral blood samples of young (left) and elderly (center) individuals and AD patients (right); (**b**) the average of CLIC1 fluorescence measured in single monocytes is not significantly different between young (*n* = 20) and old (*n* = 27) individuals, while it is significantly higher in AD patients (*n* = 25) (****, *p* < 0.0001, AD vs. young and AD vs. old, one-way ANOVA, Tukey’s multiple comparisons test).

**Figure 2 ijms-21-01484-f002:**
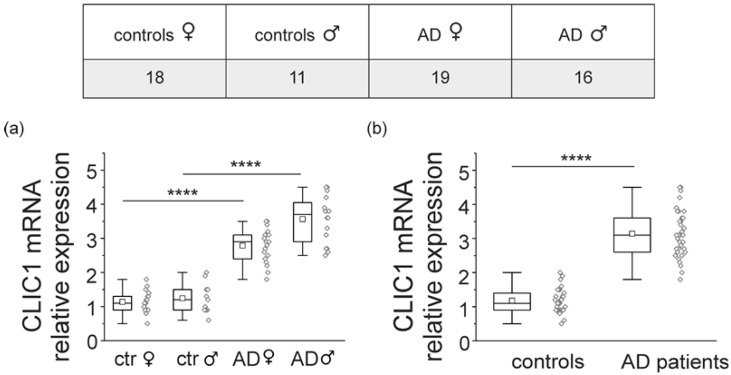
CLIC1 mRNA overexpression in peripheral blood mononuclear cells (PBMCs) from AD patients. (**a**) CLIC1 mRNA relative expression in PBMCs from 18 female and 11 male healthy controls is significantly lower compared to the CLIC1 mRNA level in peripheral blood cells from 19 female and 16 male AD patients (****, *p* < 0.0001, one-way ANOVA, Tukey’s multiple comparisons test). (**b**) All the data in (a) are pulled together and compare 29 controls individuals (1.179 ± 0.068) with 35 AD patients (3.143 ± 0.118) (****, *p* < 0.0001, unpaired *t* test).

**Figure 3 ijms-21-01484-f003:**
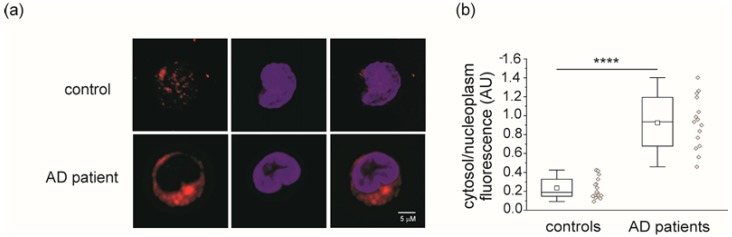
Cytoplasmic accumulation of CLIC1 protein in monocytes from AD patients. (**a**) Representative confocal images of single isolated monocytes (blue nuclei) from control subjects (top) and AD patients (bottom) showing CLIC1 protein distribution (red). (**b**) Analysis of the cytoplasm/nucleoplasm CLIC1 fluorescence ratio in isolated monocytes from 18 control individuals (0.235 ± 0.027) and 15 AD patients (0.925 ± 0.071) (****, *p* < 0.0001, unpaired *t* test).

**Figure 4 ijms-21-01484-f004:**
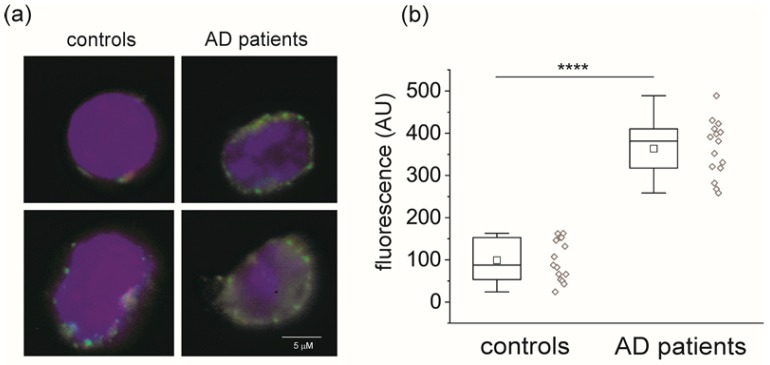
Membrane accumulation of CLIC1 protein in monocytes from AD patients. (**a**) Representative confocal images of transmembrane CLIC1 immunostaining (green) in single monocytes (blue nuclei) obtained from healthy controls (left) and AD patients (right). (**b**) Quantification of CLIC1 membrane fluorescence in single monocytes from control subjects (*n* = 15) and AD patients (*n* = 15). Transmembrane CLIC1 staining is significantly increased in AD patients (363.6 ± 17.16) compared to healthy controls (99.15 ± 12.56) (****, *p* < 0.0001, unpaired t test) (see Appendix A).

**Figure 5 ijms-21-01484-f005:**
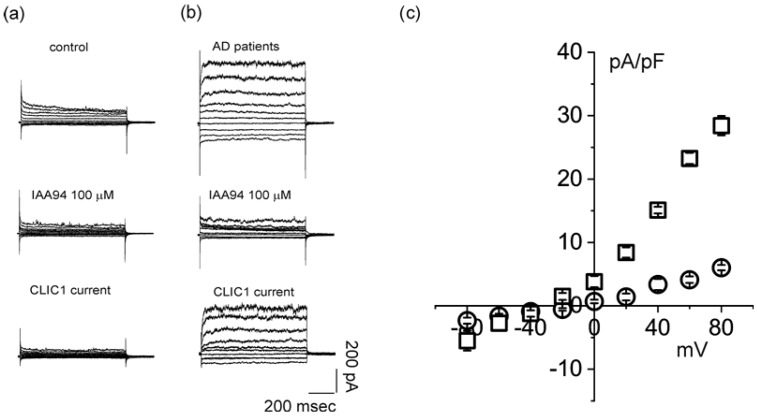
Electrophysiology measurements of isolated monocytes. Perforated patch-clamp recordings of membrane currents elicited by 800 msec voltage steps from −80 to +80 mV every 20 mV in control cells (**a**) and monocytes from AD patients (**b**). CLIC1 currents (bottom traces) have been isolated subtracting IAA94 100 µM sensitive current from total currents. (**c**) Current/voltage relationship of CLIC1 membrane current density in monocytes from AD patients (*n* = 6, open squares) and from control individuals (*n* = 5, open circles). Statistics analysis: unpaired *t*-test (*, *p* = 0.0137 at 0 mV; ****, *p* < 0.0001 at +20, +40, +60 and +80 mV).

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
