# Peer review of "CLIC1 Protein Accumulates in Circulating Monocyte Membrane during Neurodegeneration"

_ijms, 2020, doi:10.3390/ijms21041484_

Round 1

Reviewer 1 Report

The manuscript is devoted to the search of blood-based biomarkers of neurodegenerative (ND) processes using Alzheimer’s disease (AD) as an example. There is a high need in such biomarkers, although the author’s statement that “the only reliable method is the measurement of beta amyloid tau peptides concentration in the Cerebrospinal Fluid (CSF)” is not correct. Measurement of other proteins such as NFL in CSF, imaging technologies, assays based on analysis of proteins, microRNAs in plasma/serum, although less commonly used, are only some examples. Nevertheless, new blood-based biomarkers of AD and other NDs would be very useful. The authors propose to use as such a biomarker the Chloride Intracellular Channel 1 (CLIC1) protein in Peripheral Blood Mononuclear Cells (PBMCs), in particular its transmembrane form (tmCLIC1) and provide experimental data supporting their idea.

The results are interesting but the manuscript needs some changes and editing.

Although the authors discuss the significance of early ND, and AD in particular, detection, the presented data have been obtained for advanced AD, which should be clearly discussed. Young controls are presented in Fig. 1 but from Materials and Methods their source is not clear (only age- and sex-matched controls are described). It is unclear how many monocytes from each subject were tested by immunostaning, what the authors mean in the section 4.6: “as means +/- SEM of at least three independent experiments”. It is unclear why they are using SEM, not standard deviation (SD). Why in Fig. 4 SEMs are identical (12.6) for controls and AD, although means are >3 times different? Were all monocytes or only some of them similarly enriched in AD with tmCLIC1? From statistical viewpoint mRNA data look more convincing. In many cases the authors are discussing NDs, although only AD has been investigated. This should be clearly stated. If appearance of tmCLIC1 in monocytes reflects inflammatory processes, it would be reasonable to analyze some inflammatory processes, which do not involve CNS to answer the question how specific are observed effects for neuroinflammations. In any case it should be added to Discussion.

Generally speaking experiments and statistics should be described in more details and Discussion should reflect various possible explanations of data obtained.

Reviewer 2 Report

Summary

Carlini et al have reported an interesting study looking at CLIC1 expression in peripheral blood monocytes and how this could be useful in use as a blood-based biomarker for AD. The need for biomarkers to identify Adin its early stages is of critical importance in the population, and the elderly most susceptible to the development of AD and related dementias. Investigations into blood-based markers is not a new approach, but the present study focuses on CLIC1 expression and localization on circulating monocytes and how this correlates with AD. Studies like these are needed in AD and other neurodegenerative diseases, so this work is valuable. However, there are several key points described below that the authors should address to clarify and improve the manuscript.

Comments & Suggestions

Major

Recent research implicates membrane CLIC1 aggregation in cellular cytokinesis (Kagiali et al, 2020), which is a separate but important biologic process in itself. Is it possible that this function of CLIC1 could play a role in some of the observations made in the present study? As this is an important alternative reason for CLIC1 membrane aggregation, the authors should make some Discussion about how this involvement of CLIC1 may influence the interpretation of the data in the study and how to keep this in consideration for future experiments.

In Figure 3, how are cytoplasm and nucleoplasm differentiated? This is difficult to tell from the images, and the value of this relationship is called into question by the studies later experiments. Since ultimately, the paper indicates membrane CLIC1 is the most representative marker in these cells for biomarkers for AD, why not just describe CLIC1 generally as intracellular in these figures?

CLIC1 mRNA expression is generally examined in monocytes as a whole between controls and AD patients (as shown in Figure 2), and the difference is striking. This further adds questions as to the value of the examination of cytoplasmic/nucleoplasmic ratio of CLIC1 in subsequent experiments and figures. The authors even state false positives are more likely by assessing cytoplasmic/nucleoplasmic ratio compared to overall expression and localization. It seems to that intracellular vs. membrane expression is the key to this story, so it may be best to focus on this more and really explain the importance of the work related to CLIC1 expression and localization in monocytes as a marker for AD. Perhaps the characterization of the intracellular compartmentalization could be placed in supplemental data for readers to examine if they are interested in the rationale for studying the total vs. membrane expression of CLIC1 for the purposes of this study?

Since this study was in vitro characterization of CLIC1 in healthy and AD patient circulating monocytes, and monocytes are recruited into the CNS in degenerative and traumatic conditions to serve as resident macrophages, how would the authors describe the relevance of this behavior of the monocytes in the context of CLIC1 expression and localization as peripheral biomarkers on these cells? This is not well explained but is critical to the relevance and overall story being told here. Are cells altering this expression once in the CNS and then extravasate back out into the circulatory system? In general, what is the connection between the influence of monocytes and CLIC1 in AD and its use as a circulating biomarker on monocytes? Though this is very complex, some added detail on mechanisms on why the results observed in this study occur and what it is useful for evaluating these cells are important for follow up studies. The Discussion is too short and adding these details as well as others discussed above could help expand on identifying the importance of this work and issues to keep in mind for future research.

Reviewer 3 Report

In their manuscript “CLIC1 protein accumulates in circulating monocyte membrane during neurodegeneration” Carlini and colleagues use isolated peripheral blood mononuclear cells (PBMCs) from Alzheimer’s patients and age matched controls to study the quantity, distribution, and physiological activity of the Chloride intracellular channel CLIC1. CLIC1 can exist as either a soluble cytoplasmic/nucleoplasmic or an integral dimeric transmembrane channel. Conversion to the latter is induced by stress factors/inflammation. Chloride flux changes monocyte physiological activity, the authors hypothesize that this might enable them to cross the blood brain barrier to become brain resident macrophages. The authors find that CLIC1 is enriched in PBMCs isolated from Alzheimer’s patients, more specifically transmembrane CLIC1. Patch clamp recording confirmed increased chloride currents in Alzheimer’s derived PBMCs. I find their data very important, convincing and very valuable for the field. Their findings might be of relevance to find a novel biomedical marker of neuroinflammation in neurodegenerative disorders. A few adjustments and clarifications need to be done before the manuscript is ready for publication.

Major concerns

PBMCs include lymphocytes (70-90%), monocytes (10-20%), and dendritic cells (1-2%).

1.a. It seems like the authors use the definition PBMCs and monocytes interchangeably. This has to be corrected/clarified throughout the text.  

1.b. How do the authors make sure that the cells they are looking at are monocytes and not lymphocytes? Did they further purify the monocytes (FACS sorting, isolation kit), or perform a co-labeling?

1.c. Is CLIC1 expressed in lymphocytes?

P-values in figures: It is hard to believe that the significance in all figures (1-4) is of the same magnitude (p<0.0001). Please double-check and provide the correct p-values for each picture. Figure 5 does not have the p-values indicated in the graph, but refers to them in the legend. Figure 3. How was the nucleoplasm defined? DAPI? All ICC figure legends are missing the information about DAPI staining. Especially in case of figure 3 it would be important to show individual color channels, rather than just the merged picture. The authors describe and use a “custom-made primary mouse polyclonal anti-NH2 CLIC1 antibody”.

4.a. What is the exact epitope?

4.b. Where was the antibody generated (source: company, own laboratory)?

4.c. Was the antibody tested for specificity? Westernblot? The authors need to provide convincing data about the antibody’s specificity. Or a reference that provides such data.

The methods section reveals that the AD patient / control individual data includes Abeta and Tau levels. Did the authors do correlation studies to see if Tau and or Abeta levels correlate with tmCLIC1? In total there were 29 controls and 35 AD samples. In figure 2 all samples have been analyzed. However, in figure 1, 3, 4, 5, not all samples were used. Particularly figure 5 has a very low n-number (4 and 5, respectively). Thus, it is no surprise that the significance in figure 5 is very high, especially given the potential idea the authors used samples with the best response in previous experiments. Which of course would be a fair procedure, but needs to be clearly stated, if done. Due to the low n-number it is not possible to compare the precision of currency-measurement with the other methods. Respective statements in the manuscpt need to be corrected.

Minor concerns:

The abstract could briefly explain the functional relevance of transmembrane CLIC1 versus soluble. Please introduce IAA94 in the results. In the methods it is just specified as a “specific inhibitor”, change to “specific Chloride channel inhibitor”. “CLIC1 peculiar localization would be functional to monocytes proliferation and infiltration [19].” Reference 19 is Cha et al., 2007, Mol Cell Proteomics. It is not clear to me how this referenced manuscript shows/indicates/contributes to this hypothesis? The images show very patchy/polar CLIC1 distribution. Is it possible that CLIC1 follows a polar distribution pattern? Especially with regards to its function in cell division? Material and Methods: the age of patients and control individuals has to be provided Supplemental Figure 3, what is labeled in red/yellow? The introduction could briefly introduce what is known about CLIC1 function in the nucleus.

Round 2

Reviewer 2 Report

The authors have sufficiently addressed my concerns with their revisions of the manuscript.